# SARS-CoV2 variant-specific replicating RNA vaccines protect from disease following challenge with heterologous variants of concern

David W Hawman[1]*, Kimberly Meade-White[1], Jacob Archer[2], Shanna S Leventhal[1], Drew Wilson[1], Carl Shaia[3], Samantha Randall[4], Amit P Khandhar[2], Kyle Krieger[2], Tien-Ying Hsiang[5], Michael Gale[5], Peter Berglund[2], Deborah Heydenburg Fuller[4], Heinz Feldmann[1]*, Jesse H Erasmus[2,4]*

[1]Laboratory of Virology, Division of Intramural Research, National Institute of Allergy and Infectious Diseases, National Institutes of Health, Rocky Mountain Laboratories, Hamilton, United States; [2]HDT Bio, Seattle, United States; [3]Rocky Mountain Veterinary Branch, Division of Intramural Research, National Institute of Allergy and Infectious Diseases, National Institutes of Health, Rocky Mountain Laboratories, Hamilton, United States; [4]Department of Microbiology, University of Washington School of Medicine, Seattle, United States; [5]Center for Innate Immunity and Immune Disease, Department of Immunology, University of Washington School of Medicine, Seattle, United States

*For correspondence:
david.hawman@nih.gov (DWH);
feldmannh@niaid.nih.gov (HF);
jesse.erasmus@hdt.bio (JHE)

**Abstract** Despite mass public health efforts, the SARS-CoV2 pandemic continues as of late 2021 with resurgent case numbers in many parts of the world. The emergence of SARS-CoV2 variants of concern (VoCs) and evidence that existing vaccines that were designed to protect from the original strains of SARS-CoV-2 may have reduced potency for protection from infection against these VoC is driving continued development of second-generation vaccines that can protect against multiple VoC. In this report, we evaluated an alphavirus-based replicating RNA vaccine expressing Spike proteins from the original SARS-CoV-2 Alpha strain and recent VoCs delivered in vivo via a lipid inorganic nanoparticle. Vaccination of both mice and Syrian Golden hamsters showed that vaccination induced potent neutralizing titers against each homologous VoC but reduced neutralization against heterologous challenges. Vaccinated hamsters challenged with homologous SARS-CoV2 variants exhibited complete protection from infection. In addition, vaccinated hamsters challenged with heterologous SARS-CoV-2 variants exhibited significantly reduced shedding of infectious virus. Our data demonstrate that this vaccine platform can be updated to target emergent VoCs, elicits significant protective immunity against SARS-CoV2 variants and supports continued development of this platform.

## Editor's evaluation

This manuscript aims to develop second-generation vaccines that protect against multiple SARS-CoV2 variants. The overall experimental design and the data are very nice. In addition, authors reasonably revised the original manuscript.

## Introduction

Since the emergence of SARS-CoV2 in China in late 2019, over 200 million confirmed cases have been reported. The global public health burden caused by SARS-CoV2 and its resulting disease, coronavirus disease 2019 (COVID-19), has driven the rapid development of several promising vaccine candidates

**eLife digest** Since 2019, the SARS-CoV-2 virus has spread worldwide and caused hundreds of millions of cases of COVID-19. Vaccines were rapidly developed to protect people from becoming severely ill from the virus and decrease the risk of death. However, new variants – such as Alpha, Beta and Omicron – have emerged that the vaccines do not work as well against, contributing to the ongoing spread of the virus.

One way to overcome this is to create a vaccine that can be quickly and easily updated to target new variants, like the vaccine against influenza. Many of the vaccines made against COVID-19 use a new technology to introduce the RNA sequence of the spike protein on the surface of SARS-CoV-2 into our cells. Once injected, our cells use their own machinery to build the protein, or 'antigen', so the immune system can learn how to recognize and destroy the virus.

Here, Hawman et al. have renovated an RNA vaccine they made in 2020 which provides immunity against the original strain of SARS-CoV-2 in monkeys and mice. In the newer versions of the vaccine, the RNA was updated with a sequence that matches the spike protein on the Beta or Alpha variant of the virus. Both the original and updated vaccines were then administered to mice and hamsters to see how well they worked against SARS-CoV-2 infections.

The experiment showed that all three vaccines caused the animals to produce antibodies that can neutralize the original, Alpha and Beta strains of the virus. Vaccinated hamsters were then infected with one of the three variants – either matched or mismatched to their vaccination – to see how much protection each vaccine provided. All the vaccines reduced the amount of virus in the animals after infection and mitigated damage in their lungs. But animals that received a vaccine which corresponded to the SARS-CoV-2 strain they were infected with had slightly better protection.

These findings suggest that these vaccines work best when their RNA sequence matches the strain responsible for the infection; however, even non-matched vaccines still provide a decent degree of protection. Furthermore, the data demonstrate that the vaccine platform created by Hawman et al. can be easily updated to target new strains of SARS-CoV-2 that may emerge in the future.

Recently, the Beta variant of the vaccine entered clinical trials in the United States (led by HDT Bio) to evaluate whether it can be used as a booster in previously vaccinated individuals as well as unvaccinated participants.

with a few currently being used for large-scale vaccination efforts (*Subbarao, 2021*). However, the continued emergence of novel SARS-CoV2 variants of concern (VoCs) and the diminished efficacy of current vaccines against infection by these VoCs has raised concerns that novel SARS-CoV2 variants could evade current vaccine-mediated immunity (*Subbarao, 2021*). While leading mRNA vaccines provide protection from severe COVID-19 disease and hospitalization in individuals infected with VoCs (*Lopez Bernal et al., 2021*), breakthrough infections, particularly in the upper airways of vaccinated individuals, are increasing and can lead to transmission events, even within clusters of vaccinated individuals (*Brown et al., 2021*). This highlights the need for continued vaccine development activities to improve protection from infection as well as disease to reduce transmission in the population. In addition, although current vaccine candidates are widely available in developed countries, access to vaccines in less developed countries is still lacking (*Fontanet et al., 2021*). Thus, development of new vaccine candidates capable of inducing broad and protective immunity and amenable to worldwide distribution is still necessary to increase the availability of vaccines to address the pandemic and reduce the likelihood of vaccine-resistant strains of SARS-CoV2 emerging.

Previously, we reported the development of an Venezuelan equine encephalitis virus (VEEV)-based replicating RNA (repRNA) vaccine encoding the spike protein of the original A.1 lineage of SARS-CoV2 delivered by a lipid inorganic nanoparticle (LION) (*Erasmus et al., 2020a*). We have since transitioned this technology through current good manufacturing practices-compliant production of both the repRNA and LION components and demonstrated safety and tolerability in a preclinical toxicology study leading to our open investigational new drug status under the drug product name, HDT-301, a B.1.351-specific vaccine that is pending phase I trials in US. Additionally, this technology is currently being evaluated in a Phase I clinical trial in India under the drug product designation HGC019 with pending filings in Brazil, South Korea, and China.

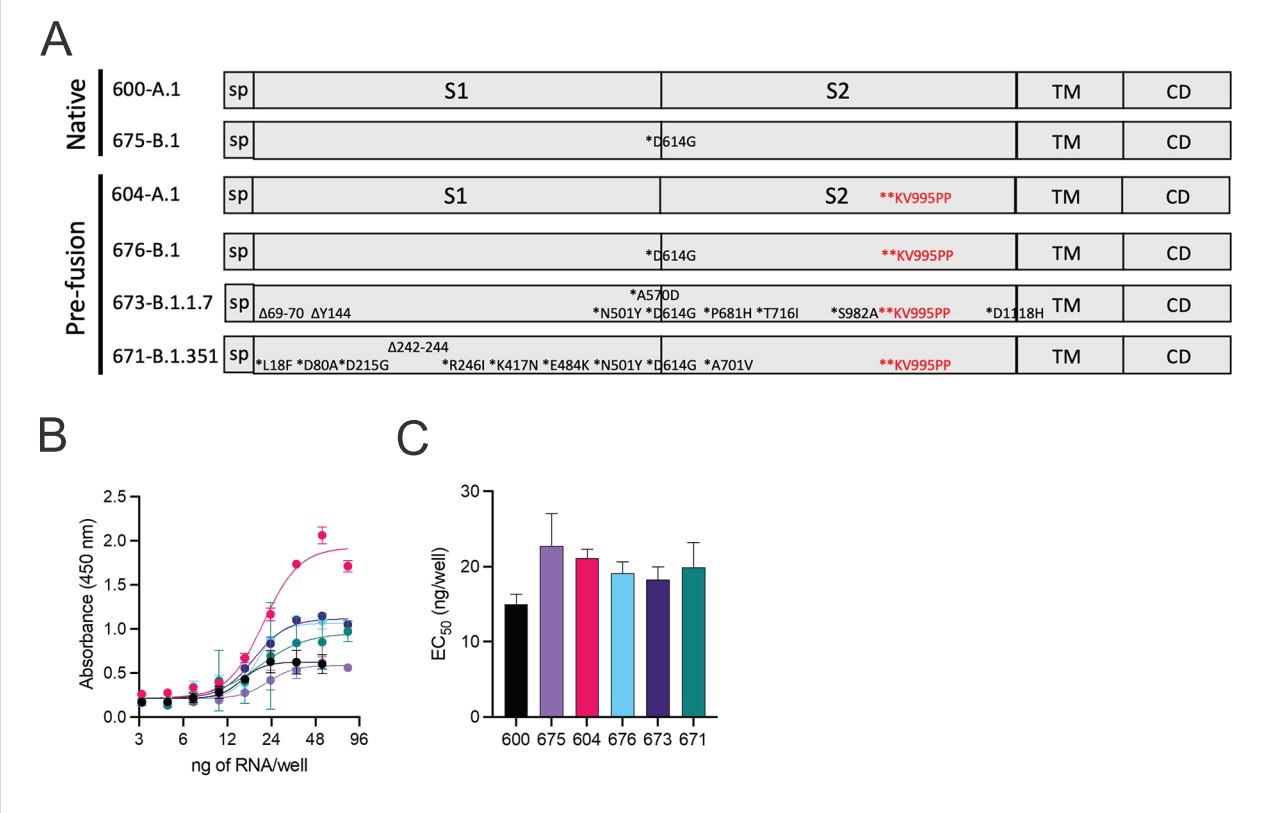

**Figure 1.** Design of full-length SARS-CoV2 spike immunogens harboring deletions and/or substitutions specific to variants of concern. (**A**) Using the full-length, pre-fusion-stabilized (KV995PP) spike (**S**) of the original A.1 lineage of SARS-CoV2 as a reference, including the signal peptide (sp) and regions corresponding to S1, S2, transmembrane (TM), and cytoplasmic (CD) domains, the various deletions and/or substitutions corresponding to the B.1, B.1.1.7, and B.1.351 lineages were introduced. Additionally, a subset of constructs were prepared on the native version of spike. These six open reading frames were cloned into the replicating RNA (repRNA) backbone downstream of the sub-genomic promoter prior to production of capped mRNA. (**B**) RNA of each construct was then formulated with lipid inorganic nanoparticle (LION) and serial dilutions added to monolayers of baby hamster kidney cells. Cell surface-bound spike was then measured by sandwich enzyme linked immunosorbent assay (ELISA) of cell lysates harvested at 24 hr post-transfection and (**C**) half-maximal effective concentrations ($EC_{50}$) determined for each RNA construct.

The online version of this article includes the following source data for figure 1:

**Source data 1.** Source data for *Figure 1*.

In this report we evaluated repRNA vaccine expressing the SARS-CoV2 spike protein of the SARS-CoV2 variants, B.1.1.7 and B.1.351 in mice and hamsters, using LION to mediate delivery via intramuscular (IM) injection. Vaccination of mice or hamsters with LION/repRNA vaccines expressing spikes of distinct VoCs induced potent neutralizing antibody (nAb) responses against each virus after a prime-boost regimen. Vaccinated hamsters challenged with homologous SARS-CoV2 strains were completely protected from infection. In addition, vaccinated hamsters challenged with heterologous viruses were significantly protected against viral shedding, viral replication, and exhibited little-to-no viral-induced pathology in the lung tissue. Cumulatively, our data provide direct evidence that a repRNA/LION vaccine can provide complete protection from exposure to homologous viruses and substantial protection from disease via accelerated clearance of virus when exposed to a new heterologous VoC. These data support further development of the LION/repRNA vaccine.

## Results

### Design and production of VoC-targeted repRNA-CoV2S

Using sequences deposited on GISAID, we designed variant spike open reading frames described in *Figure 1A* for cloning into our previously described repRNA-CoV2S (*Erasmus et al., 2020a*). Following synthesis and cloning, we sequence-verified each construct prior to production of RNA for

in vitro qualification using an in vitro potency assay that utilizes a sandwich enzyme linked immuno-sorbent assay (ELISA) to quantify cell surface-expressed SARS-CoV2 spike (*Figure 1B and C*). While differences in cell surface spike expression were observed at higher concentrations of LION/repRNA (*Figure 1B*), the relative potencies, in terms of half-maximal effective concentration (EC$_{50}$), were not significantly different from each other, ranging from 15 to 21 ng of repRNA per well. The differences at higher concentrations could either be due to variability in antibody binding or related to the mutations present in each spike, including the KV995PP mutation which was associated with an increased signal at the higher concentrations (optical densities of 0.6–2.1 for constructs 600 vs. 604, and 0.6–1.0 for constructs 675 vs. 676). To determine whether this apparent enhancement of cell surface-expressed spike, conferred by the KV995PP mutation, can contribute to enhanced immunogenicity, either in terms of magnitude and/or breadth, we included the non-2P-stabilized as well as the 2P-stablized spike of the B.1 spike in the subsequent mouse immunogenicity study, as well as the 2P-stabilized B.1.1.7 and B.1.351 spikes.

## Variant-specific immunization provides distinct binding and cross-neutralization profiles

To evaluate homologous and heterologous neutralization following prime and boost vaccination, we immunized mice with LION/repRNA expressing the SARS-CoV2 spike from several VoC, including the native spikes of A.1 or B.1 lineages as well as the pre-fusion spikes of B.1, B.1.1.7, or B.1.351 lineage viruses. Mice were vaccinated with 1 µg of RNA and boosted with the identical immunogens 28 days later. At 14 days post-boost, serum samples were assayed by ELISA for binding to the A.1 spike (S) or its S1, S2, or receptor-binding (RBD) domains (*Figure 2A*). Samples were also assayed by 80% plaque reduction neutralization test (PRNT$_{80}$) against A.1, B.1, B.1.1.7, or B.1.351 viruses (*Figure 2B–F*).

In terms of the binding antibody characteristics, similar binding responses to the full-length S or to the S2 domain and RBD were observed between vaccine groups (*Figure 2A*), however the B.1.1.7- or B.1.351-vaccinated mice appeared to mount higher responses to the S1 domain with significantly higher S1-binding responses in the B.1.1.7-vaccinated mice compared to the ancestral A.1-vaccinated group. In terms of neutralization, in native A.1 spike-vaccinated mice, we observed similar neutralization activity against B.1 challenge virus compared to homologous A.1 challenge, while we found a significant 4.4-fold and 14.7-fold drop in neutralization activity against B.1.1.7 and B.1.351 challenge virus, respectively (*Figure 2B*). In native- or pre-fusion B.1-spike-vaccinated mice, we observed similar magnitudes of neutralizing responses against A.1, B.1, and B.1.1.7 viruses between the two vaccines (*Figure 2C and D*), however the pre-fusion-stabilized B.1 spike appeared to induce more breadth in specificity with only a 5.7-fold drop in neutralizing activity against B.1.351 virus (*Figure 2D*), compared to a 29.3-fold drop in neutralizing activity against B.1.351 virus in mice that received the native conformation of the B.1 spike (*Figure 2C*). In pre-fusion B.1.1.7-vaccinated mice, we found a significant 11.3-fold drop in neutralization activity against the B.1.351 VoC compared to homologous B.1.1.7 virus (*Figure 2C*). In B.1.351-vaccinated mice, we found a significant 12.3-fold and 4-fold drop in neutralization activity against the B.1.1.7 and A.1 challenge virus, respectively (*Figure 2D*). Cumulatively, our data show that the B.1.351 VoC was the most resistant to nAbs elicited by A.1, B.1, or B.1.1.7 vaccination.

## Vaccination induces homologous and heterologous neutralizing antibodies in hamsters

Given the distinct nAb profiles of the B.1.1.7 and B.1.351 vaccines, relative to each other, we next evaluated these two candidates, as well as a reference A.1 lineage spike, on the same pre-fusion-stabilized spike backbone, in the Syrian Golden hamster model of SARS-CoV2 infection. Hamsters were vaccinated with 20 µg of LION/repRNA via IM injections and boosted with identical immunizations 4 weeks later (*Figure 3A*). To evaluate immunogenicity of LION/repRNA in hamsters, at day 14 post-boost, serum samples were assayed by PRNT$_{80}$ against A.1, B.1.351, or B.1.1.7 viruses as well as the more recently described B.1.617.2 virus (Delta) variant. Against homologous virus, each vaccine candidate elicited robust nAb titers ranging from 1:320 to 1:10,240 with 2- to 14-fold reductions when measured against heterologous virus (*Figure 3B*). As observed in mice, the most 'vaccine-resistant' VoC was the B.1.351 strain, where 13- to 14-fold reductions in nAb titers were observed in hamsters immunized with A.1- or B.1.1.7-specific vaccines, respectively (*Figure 3B*). The A.1 and B.1.1.7 strains

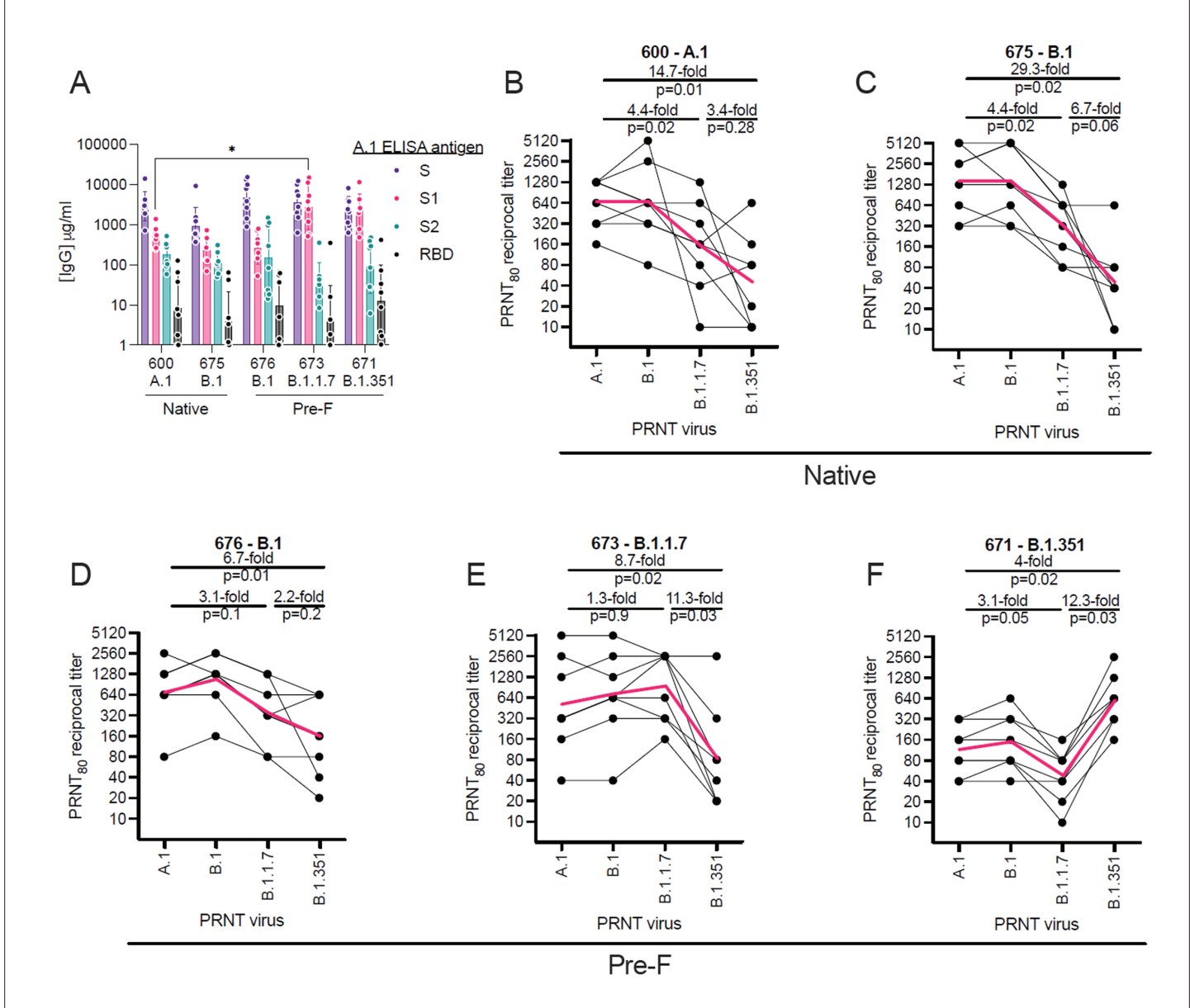

**Figure 2.** Relative binding and neutralizing antibody responses induced by each vaccine candidate. C57BL/6 mice (n = 8/group) received a 1 µg intramuscular injection on days 0 and 28 of lipid inorganic nanoparticle (LION)/replicating RNA (repRNA) encoding either the native conformation of spike derived from A.1, or B.1 viruses, or the pre-fusion-stabilized conformation of spike derived from B.1, B.1.1.7, or B.1.351 viruses, corresponding to repRNAs 600, 675, 676, 673, and 671, respectively. Mice were bled 14 days after the boost immunization and (**A**) A.1 spike (**S**)-, S1 domain-, S2 domain-, or receptor-binding domain (RBD)-binding antibody responses measured by enzyme linked immunosorbent assay (ELISA). Neutralizing antibody responses (**B–F**) measured by 80% plaque reduction neutralization tests ($PRNT_{80}$) against A.1, B.1, B.1.1.7, or B.1.351 viruses in samples from mice vaccinated with native A.1 (**B**), native B.1 (**C**), pre-fusion B.1 (**D**), pre-fusion B.1.1.7 (**E**), or pre-fusion B.1.351 (**F**) spike-derived vaccines. Data in A are presented as geometric means (±geometric standard deviations) along with each individual sample and differences in S-, S1-, S2-, or RBD-binding titers between the 600 A.1 group and other groups were compared by two-way ANOVA (*$p < 0.05$). Data in B–F are presented as each individual sample connected by black lines with the geometric mean depicted in red and differences between geometric means were compared by Student's t test.

The online version of this article includes the following source data for figure 2:

**Source data 1.** Source data for *Figure 2*.

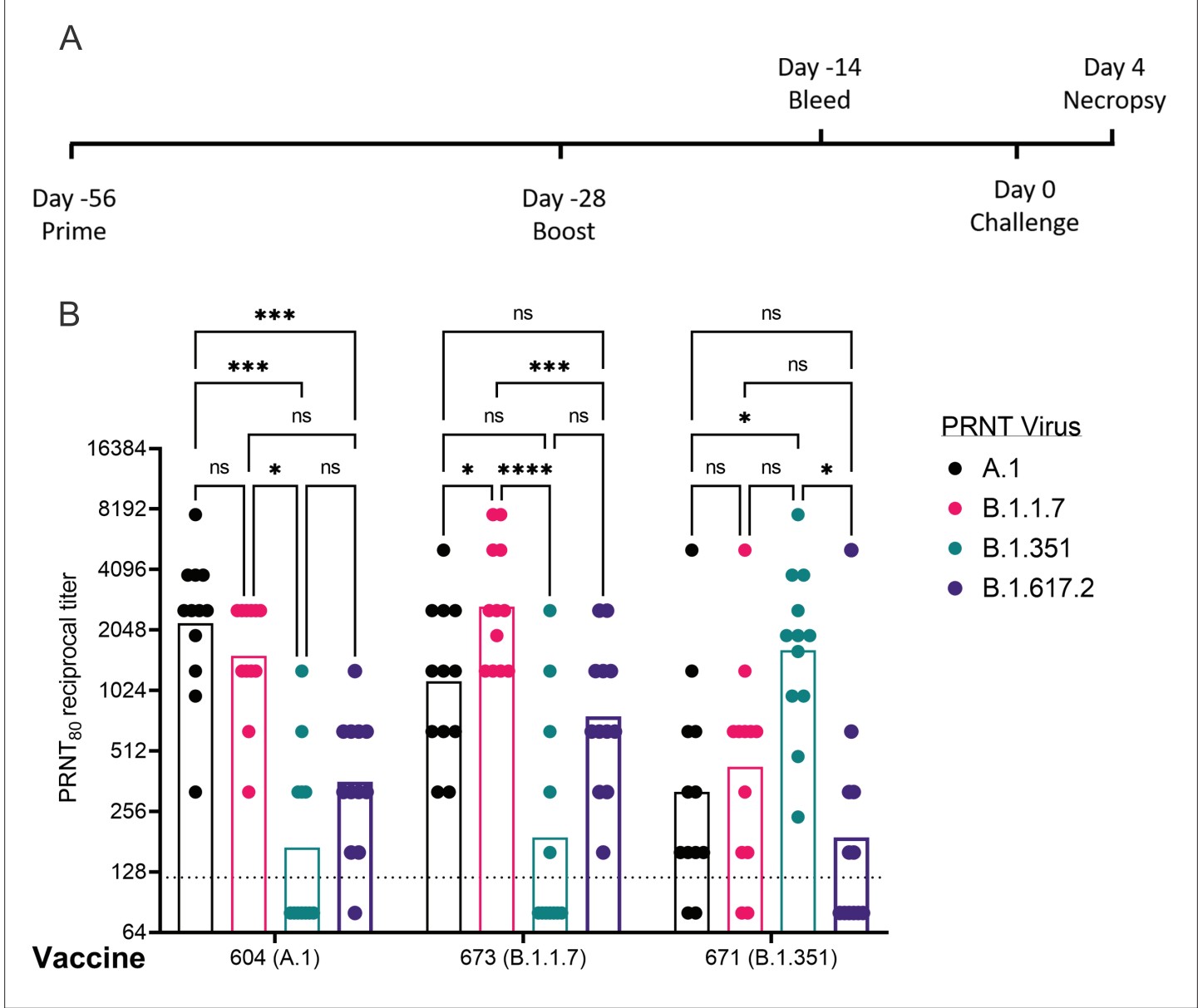

**Figure 3.** Post-boost neutralizing antibody responses in Syrian Golden hamsters. Sera from animals immunized with lipid inorganic nanoparticle (LION) complexed with replicating RNA (repRNA) vaccine variants A.1, B.1.1.7, or B.1.351 were incubated with live virus of variant A.1 (black), B.1.1.7 (pink), B.1.351 (green), or B.1.617.2 (purple) as indicated. Plaque-reduction neutralizing titers are indicated by individual symbols with geometric mean titers represented by the height of the bars. Indicated statistical comparisons performed using a two-way ANOVA with Tukey's multiple comparisons test. ns $p > 0.05$, $*p < 0.05$, $***p < 0.001$, $****p < 0.0001$.

The online version of this article includes the following source data for figure 3:

**Source data 1.** Source data for *Figure 3*.

of SARS-CoV2 were similarly neutralized by A.1 or B.1.1.7 vaccination exhibiting a roughly 2-fold reduction in neutralizing activity when comparing homologous to heterologous challenge, however, an ~4- to 5-fold reduction in nAb titer was observed against these viruses in hamsters receiving the B.1.351-specific vaccine (*Figure 3*), recapitulating observations in mice (*Figure 2*). Against the B.1.617.2 VoC, the A.1 and B.1.1.7 spike-vaccinated hamsters had a 4- to 6-fold reduction in serum neutralization titer compared to homologous virus challenge while hamsters vaccinated with the B.1.351 spike had a 10-fold reduction in neutralization titer (*Figure 3B*).

**Table 1.** Fraction of hamster samples with any detectable infectious virus.

| Challenge | RNA | Oral swabs | | Lung tissue |
| | | D2 | D4 | D4 |
| --- | --- | --- | --- | --- |
| | Mock | 6/6 | 6/6 | 6/6 |
| | A.1 (604) | 6/6 | 2/6 | 1/6 |
| | B.1.351 (671) | 6/6 | 6/6 | 1/6 |
| A.1 | B.1.17 (673) | 3/6 | 0/6 | 0/6 |
| | Mock | 5/5 | 5/5 | 5/5 |
| | A.1 (604) | 6/6 | 2/6 | 0/6 |
| | B.1.351 (671) | 2/6 | 2/6 | 0/6 |
| B.1.351 | B.1.17 (673) | 3/6 | 2/6 | 0/6 |
| | Mock | 6/6 | 6/6 | 6/6 |
| | A.1 (604) | 6/6 | 2/6 | 0/6 |
| | B.1.351 (671) | 2/6 | 3/6 | 0/6 |
| B.1.1.7 | B.1.17 (673) | 1/6 | 3/6 | 0/6 |

## Vaccination significantly reduced viral shedding

To determine the protective benefit of the candidate vaccines, hamsters were challenged with 1000 tissue-culture infectious dose 50 assay (TCID$_{50}$) of the A.1 strain or the B.1.351 or B.1.1.7 VoC 4 weeks after boosting (*Figure 3A*). Thus, all RNA vaccines would be evaluated against homologous and heterologous SARS-CoV2 challenge. For reference, the median infectious dose (ID$_{50}$) in hamsters for the A.1 strain is 5 TCID$_{50}$ (*Rosenke et al., 2020*). A significant criterion of an effective SARS-CoV2 vaccine is an ability to reduce transmission of the virus from vaccinated individuals to susceptible persons by protecting the upper airway from infection. To evaluate whether vaccination could reduce viral shedding in the upper airway of SARS-CoV2 challenged hamsters oral swabs were collected on days 2 and 4 post-infection (PI) and viral loads quantified by qRT-PCR to measure sub-genomic (Sg) RNA indicative of active viral replication and infectious virus by TCID$_{50}$ assay. In addition, we also reported samples with any detectable infectious virus, even when <1 TCID$_{50}$ to distinguish animals with no infectious virus from those with low levels of infectious virus (*Table 1*).

Against the A.1 strain, only vaccination with the B.1.1.7 repRNA significantly reduced Sg RNA levels (*Figure 4A*). It is unclear why we only observed significant reduction of A.1 viral RNA in the swabs of hamsters vaccinated with the B.1.1.7 repRNA and not the A.1 repRNA but we found non-significant differences in neutralizing activity against the A.1 strain in hamsters vaccinated with either the A.1- or B.1.1.7-specific repRNA (*Figure 3*) consistent with significant cross-protection. Additionally, the B.1.1.7-vaccinated mice did drive significantly higher A.1 S1-binding antibody responses compared to the A.1-vaccinated animals (*Figure 2A*), correlating with the significant reduction in Sg RNA levels in the oral swabs. However, all three repRNAs significantly reduced shedding of infectious virus in the oral cavity at day 2 PI and both the A.1 and B.1.1.7 repRNAs significantly reduced infectious virus at day 4 PI (*Figure 4B*). Notably, no infectious virus could be isolated from the swabs of B.1.17 repRNA-vaccinated animals at day 4 PI (*Table 1*), indicating rapid control of viral shedding. These data indicate that against the SARS-CoV2 A.1 strain, vaccination with any of the three repRNAs can significantly reduce viral shedding.

Against the B.1.351 variant, compared to mock-vaccinated hamsters at day 2 PI, hamsters vaccinated with the homologous B.1.351 repRNA had significantly reduced Sg RNA (*Figure 4C*) and by day 4 PI, significantly reduced viral RNA was seen in all vaccinated hamsters (*Figure 4C*). Similar to the A.1 challenge, against the B.1.351 variant challenge, vaccination with any of the repRNAs significantly reduced infectious virus at both day 2 and 4 PI (*Figure 4D*). Notably, at day 2 PI, infectious virus could only be isolated from the oral cavity of 2 of 6 hamsters vaccinated with the B.1.351 repRNA, while virus was isolated from 6 of 6 vaccinated with the A.1 or 3 of 6 vaccinated with the B.1.1.7 vaccine (*Table 1*), suggesting more rapid clearance of virus in B.1.351-infected hamsters vaccinated with B.1.351 repRNA.

Finally, against the B.1.1.7 variant, the B.1.351 and B.1.1.7 vaccine repRNAs significantly reduced Sg RNA at day 2 PI, and at day 4 PI both the A.1 and B.1.351 repRNAs significantly reduced Sg RNA (*Figure 4E*). Similar to A.1 or B.1.351 variant challenges, vaccination with any of the repRNAs lead to significantly reduced titers of infectious virus in oral swabs at both day 2 and 4 PI (*Figure 4F*), and more rapid clearance of virus was observed in vaccine-matched B.1.1.7-vaccinated/challenged animals, with live virus isolated in oral cavities only 1 of 6 B.1.1.7-vaccinated hamsters. In contrast, live virus could be isolated in 6 of 6 A.1- or 2 of 6 B.1.351-vaccinated hamsters challenged with the

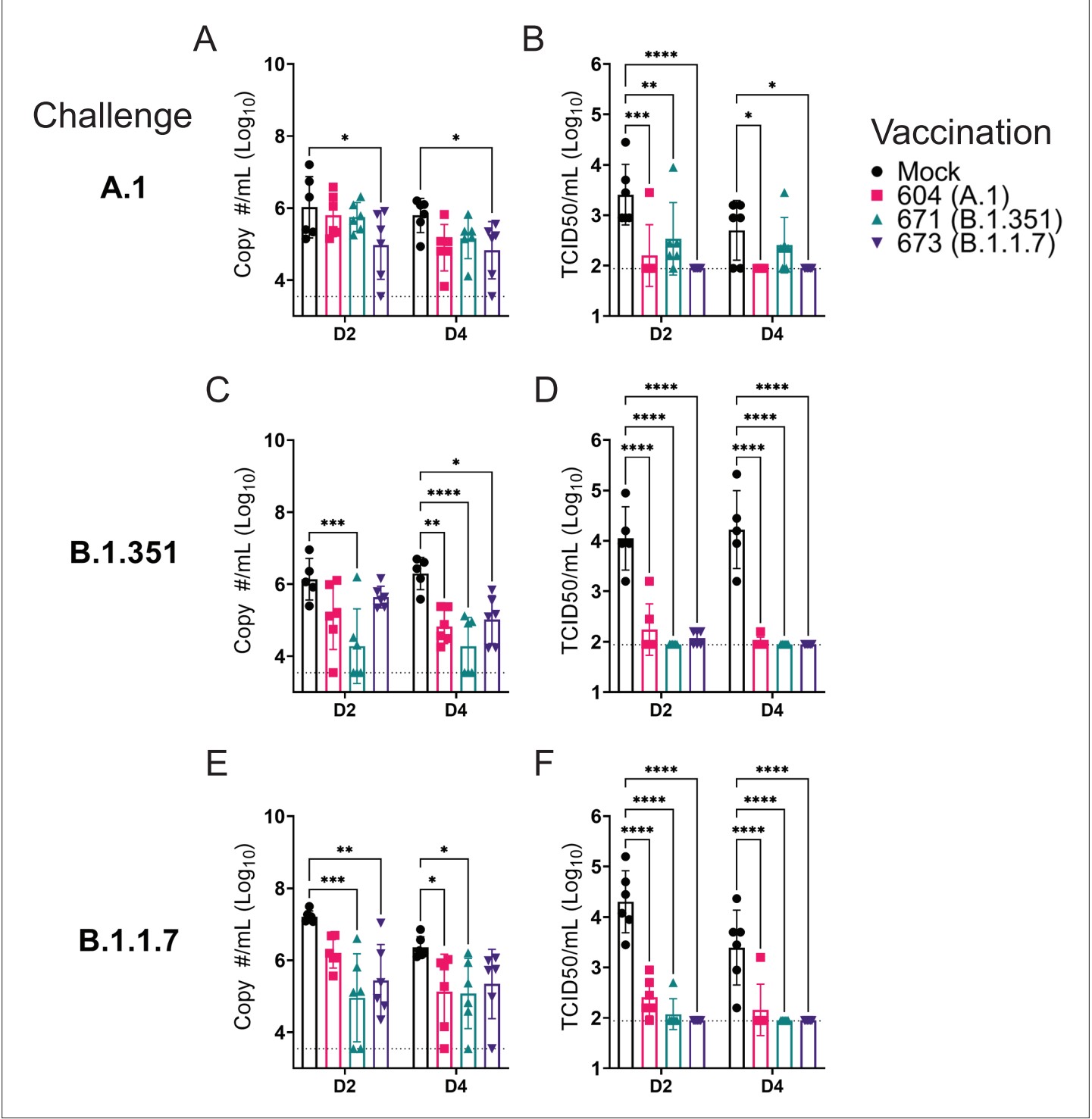

**Figure 4.** Replicating RNA (repRNA) vaccination significantly reduces viral shedding. Mock or repRNA-vaccinated hamsters were challenged with 1000 tissue-culture infectious dose 50 assay ($TCID_{50}$) of the indicated SARS-CoV2 strains via the IN route. At day 2 or 4 post-infection (PI), oral swabs were collected. SARS-CoV2 in the swabs was quantified by qRT-PCR specific for the sub-genomic (SgE) RNA (**A, C, E**) or infectious virus by $TCID_{50}$ assay (**B, D, F**). N = 5 (mock-vaccinated, B.1.351 challenge) or 6 (all other groups). A two-way ANOVA with Dunnett's multiple comparison test against mock-vaccinated hamsters was performed. *p < 0.05, **p < 0.01, ***p < 0.001, ****p < 0.0001. Comparisons without indicated p-values were considered non-significant (p > 0.05).

The online version of this article includes the following source data for figure 4:

**Source data 1.** Source data for **Figure 4**.

B.1.1.7 VoC at day 2 PI (*Table 1*). Overall, across all three SARS-CoV2 challenges, 19 of 54 and 32 of 54 vaccinated hamsters had no detectable infectious virus in their swabs at day 2 and 4 PI, respectively (*Table 1*). Cumulatively, these data demonstrate that vaccination with any of the three repRNAs significantly reduced viral shedding against homologous and heterologous SARS-CoV2 challenge with more rapid clearance observed in vaccine-matched VoC challenges.

## Vaccination significantly reduced viral burden in lung tissue

As the hamster model of SARS-CoV2 infection is not lethal, we performed a timed necropsy on day 4 PI to evaluate viral burdens in the lungs of infected hamsters via qRT-PCR and measured infectious virus in the lungs by $TCID_{50}$. All three vaccine repRNAs significantly reduced Sg viral RNA in the lungs of hamsters challenged with any of the three variants of SARS-CoV2 (*Figure 5A, C and E*). Notably, the majority of vaccinated animals across the three SARS-CoV2 variant challenges had no detectable Sg RNA, suggestive of substantial immunity in the lungs of these animals. We also quantified the amount of infectious virus in the lungs and found that all three RNA vaccines significantly reduced infectious virus in animals challenged with any of the three SARS-CoV2 variants (*Figure 5B, D and F*). Against infection with the B.1.351 and B.1.1.7 variants, no infectious virus was detected in the lungs of any vaccinated hamster (*Figure 5B, D and F* and *Table 1*). Against the A.1 strain, infectious virus was only found in the lungs of 1 of 6 animals in the A.1 and B.1.351 repRNA-vaccinated animals and 0 of 6 animals in the B.1.1.7 repRNA-vaccinated animals (*Table 1*). Cumulatively, by highly sensitive qRT-PCR for Sg RNA and by measurement of infectious virus, vaccination with any of the repRNAs resulted in significant reduction of SARS-CoV2 burden in the lungs indicating that the LION/repRNA vaccine platform is highly protective against SARS-CoV2 infection in the lung.

## Vaccination protected against lung pathology

We further evaluated vaccine efficacy by evaluating lung pathology by H&E staining, immunohistochemistry (IHC) for viral antigen, and by lung-to-body weight ratio (*Figure 6*). H&E staining of lung sections demonstrated that among mock-vaccinated hamsters challenged with A.1 or B.1.1.7 strains of SARS-CoV2 developed lesions typical of SARS-CoV2 with the A.1 and B.1.1.7 variant-infected hamsters developing more, and more severe, lesions than those infected with the B.1.351 variant. The lesions were multifocal and consisted of inflammatory cells, mostly viable and degenerate neutrophils, filling alveoli and alveolar septa were thickened by edema, fibrin, alveolar and septal macrophages, and variable amounts of type II pneumocyte hyperplasia. Bronchiolitis was present in most hamsters as was vasculitis in, or adjacent to, affected areas (*Figure 6*). In contrast, hamsters vaccinated with any of the repRNAs were protected from lung pathology and had diminished viral antigen (*Figure 6*). The majority of hamsters had no evident lesions and no detectable viral antigen and in the few that had lesions or IHC reactivity, they were minor (*Figure 6*). The complete histological and IHC findings are provided in *Supplementary file 1*.

To quantify lung pathology, we measured lung-to-body weight ratio, expressed as % of body weight and also had the H&E and IHC sections scored by a pathologist blinded to study groups. Hamsters vaccinated with any of the three repRNAs and challenged with either the A.1 or B.1.1.7 variant had significantly reduced lung-to-body weight ratios compared to similarly challenged mock-vaccinated hamsters (*Figure 7A and C*). Against the B.1.351 variant, only hamsters vaccinated with the A.1 repRNA had significantly reduced lung-to-body weight ratio compared to mock-vaccinated hamsters (*Figure 7B*). Cumulative pathology scores of H&E stained (*Figure 7D-F*) or lung sections stained for viral antigen (*Figure 7G-I*) showed that all vaccines conferred significant protection against lung pathology and significantly reduced viral antigen induced by any of the SARS-CoV2 challenge strains. Overall mock-vaccinated B.1.351 challenged hamsters had reduced viral loads (*Figure 5*) and reduced evidence of virally induced pathology (*Figure 7B, E and H*) than mock-vaccinated hamsters challenged with either the A.1 or B.1.1.7 VoC suggesting B.1.351 infection was milder in these hamsters.

## Discussion

Despite the availability of several vaccines for SARS-CoV2 in the majority of the developed world, as of early 2022, the COVID-19 pandemic continues with resurgent case numbers in multiple countries despite widespread vaccine deployment (*WHO, 2021*). The emergence of the Delta VoC, more

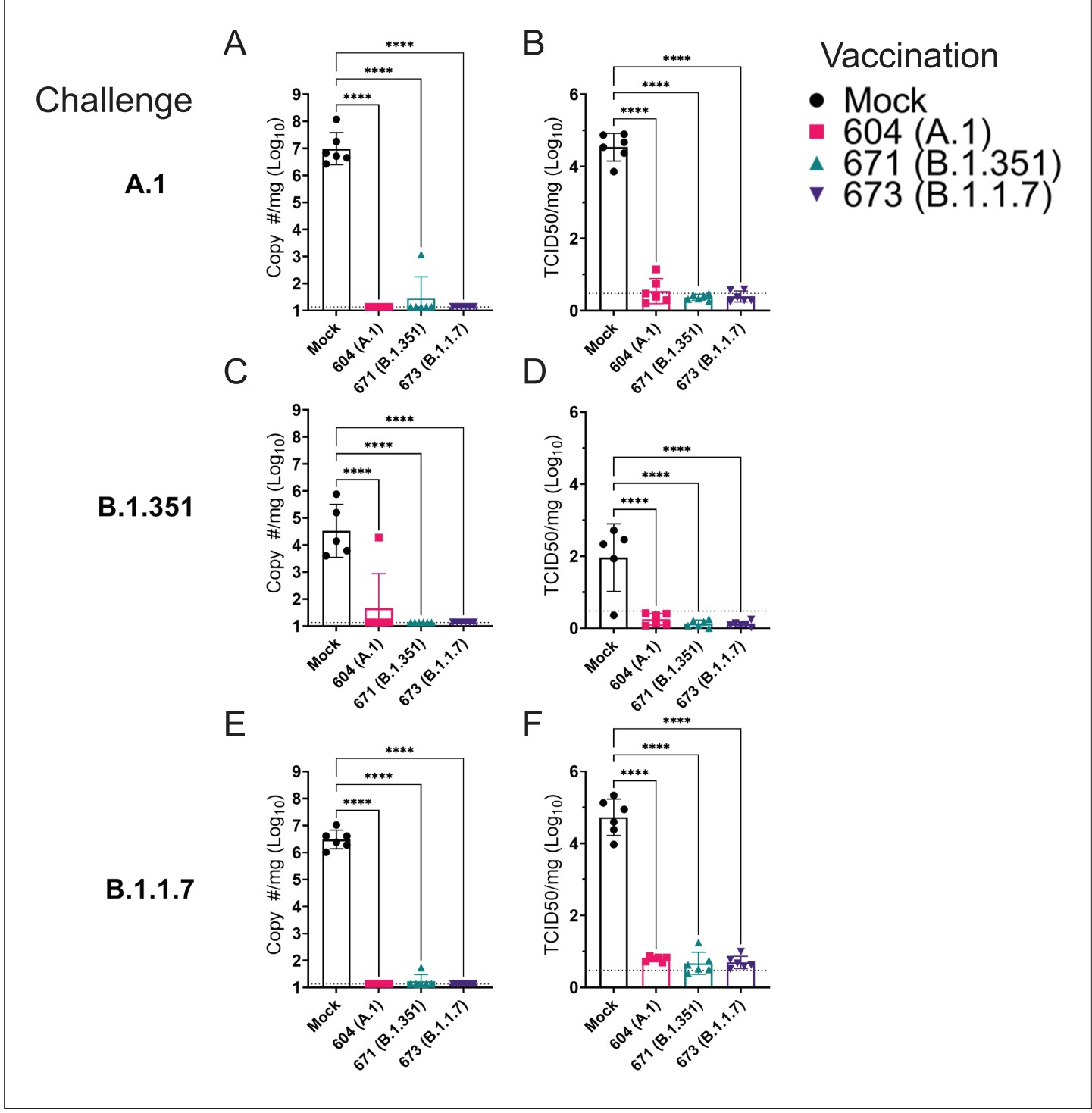

**Figure 5.** Replicating RNA (repRNA) vaccination significantly reduces viral burden in the lungs. Mock or repRNA-vaccinated hamsters were challenged with 1000 tissue-culture infectious dose 50 assay ($TCID_{50}$) of the indicated SARS-CoV2 strains via the IN route. At day 4 post-infection (PI), hamsters were euthanized, and lung tissue collected. SARS-CoV2 burden in the lung was quantified by qRT-PCR specific for the sub-genomic (SgE) RNA (**A, C, E**). Infectious virus in the lungs was quantified by $TCID_{50}$ assay (**B, D, F**). Indicated statistical comparisons performed using a one-way ANOVA with Dunnett's multiple comparison test against mock-vaccinated hamsters. *p < 0.05, ****p < 0.0001. Comparisons without indicated p-values were non-significant (p > 0.05).

The online version of this article includes the following source data for figure 5:

**Source data 1.** Source data for *Figure 5*.

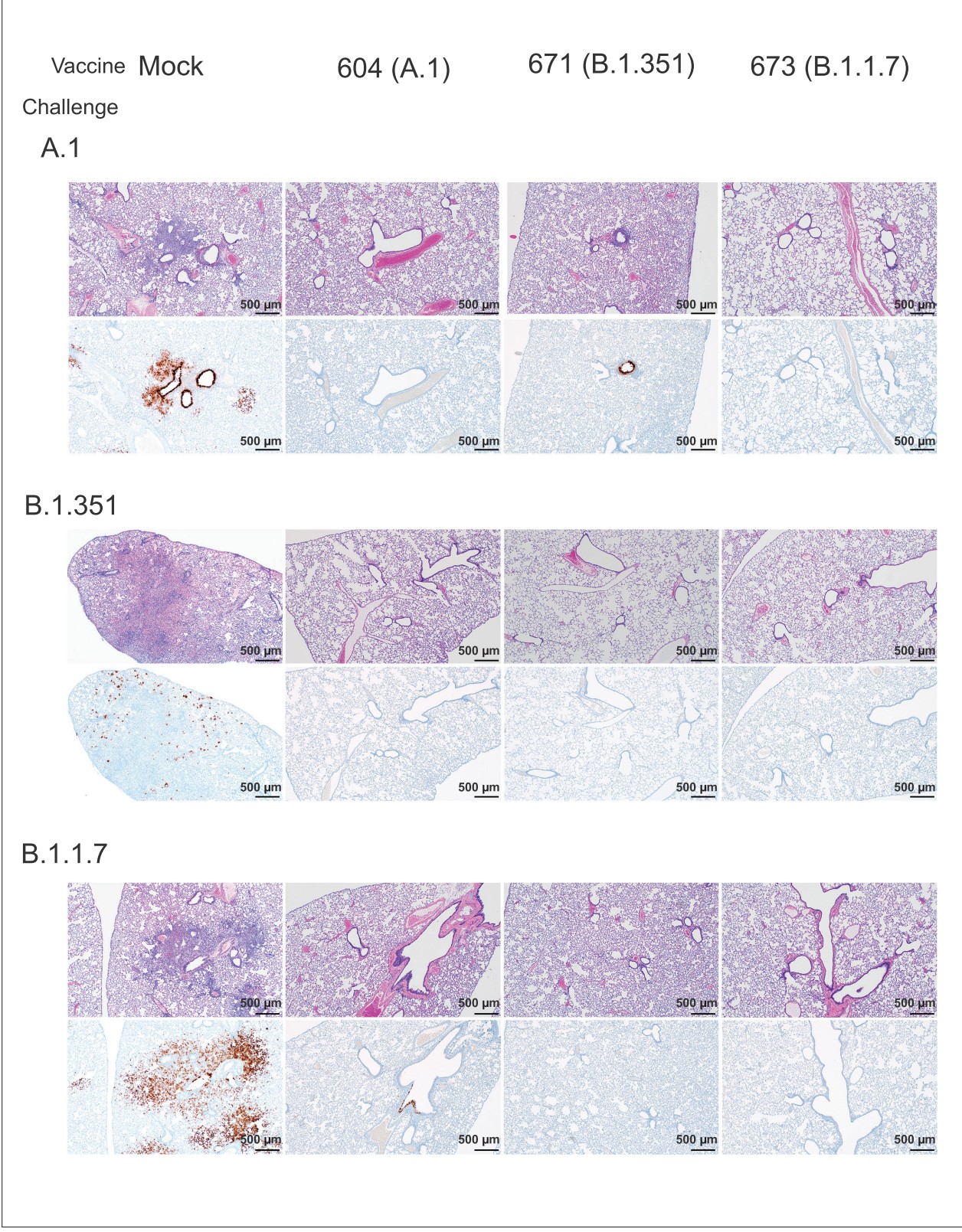

**Figure 6.** Replicating RNA (repRNA) vaccination reduces lung pathology and SARS-CoV2 antigen burden in lung tissue. Mock or repRNA-vaccinated hamsters were challenged with 1000 tissue-culture infectious dose 50 assay (TCID$_{50}$) of the indicated SARS-CoV2 strains via the IN route. At day 4 post-infection (PI), hamsters were euthanized and lungs formalin-fixed and paraffin-embedded. Sections were stained with H&E (top row of each challenge) or for SARS-CoV2 viral antigen (bottom row of each challenge). Representative images of each group are shown.

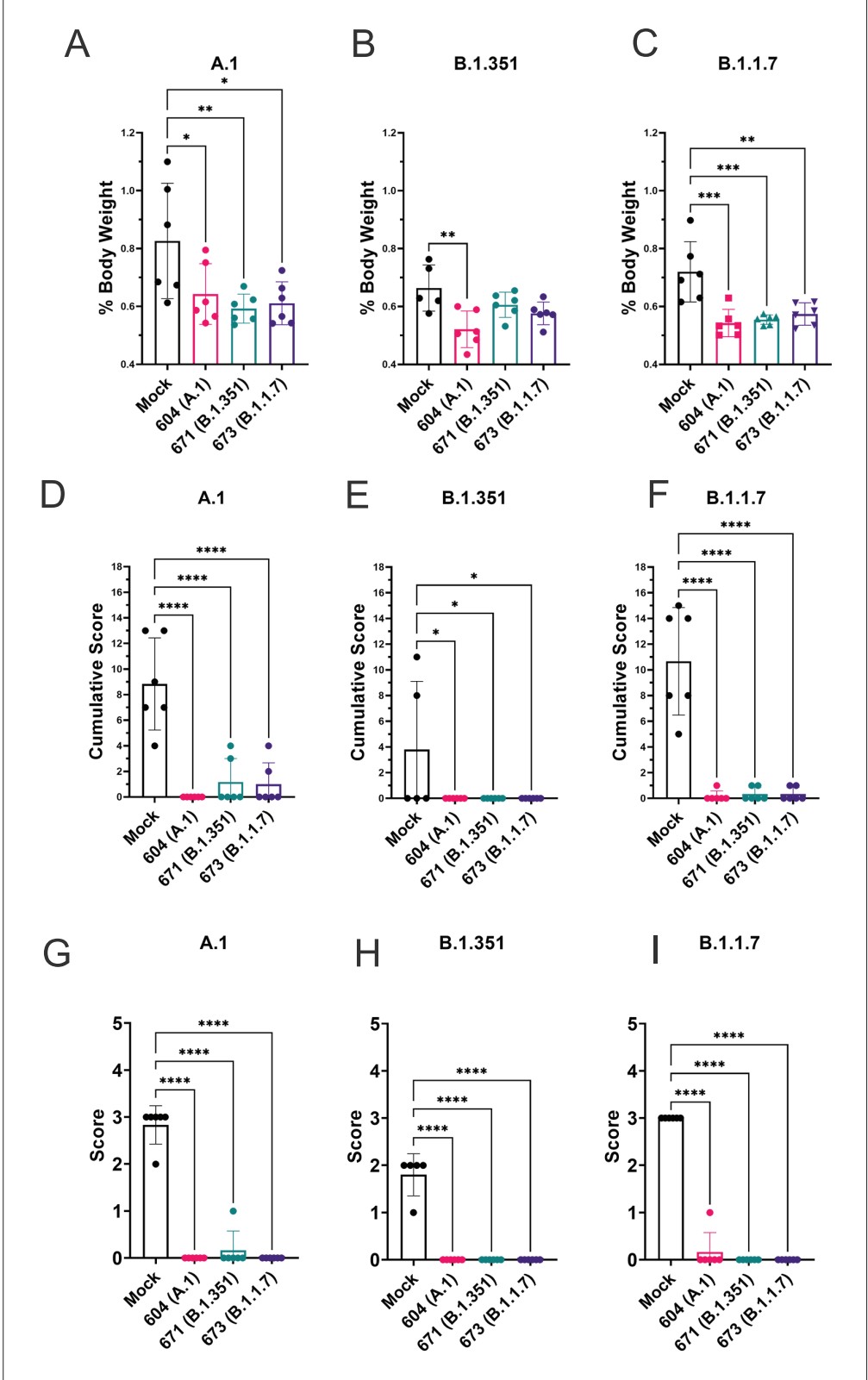

**Figure 7.** Replicating RNA (repRNA) vaccination significantly protects against lung pathology in hamsters. Mock or repRNA-vaccinated hamsters were challenged with 1000 tissue-culture infectious dose 50 assay (TCID$_{50}$) of the indicated SARS-CoV2 strains via the IN route. At day 4 post-infection (PI), hamsters were euthanized, lung weighed and lungs formalin-fixed and paraffin-embedded. Lung weights as percentage of body weight are reported

*Figure 7 continued on next page*

*Figure 7 continued*

(A–C). Lung sections were stained with H&E (D–F) or for SARS-CoV2 viral antigen (G–I). Sections were scored by a pathologist blind to study groups and assigned a score for percent area affected by SARS-CoV2 lesions and cumulative score presented (A–C) or presence of viral antigen (D–F). Indicated statistical comparisons performed using a one-way ANOVA with Dunnett's multiple comparison test against mock-vaccinated hamsters. *p < 0.05, **p < 0.01, ***p < 0.001, ****p < 0.0001. Comparisons without indicated p-values were considered non-significant (p > 0.05).

The online version of this article includes the following source data for figure 7:

**Source data 1.** Source data for *Figure 7*.

recently Omicron (B.1.1.529), and emerging evidence that existing vaccines have diminished efficacy against these VoCs compared to previous VoC (*Ai et al., 2022*; *Carreño et al., 2022*; *Lopez Bernal et al., 2021*; *Sheikh et al., 2021*; *Brown et al., 2021*) together indicate that continued development of vaccines for COVID-19 is warranted.

Several COVID-19 alphavirus-based replicon vaccines have been evaluated in animal models. In mice vaccinated with a lipid nanoparticle delivered VEEV repRNA expressing the pre-fusion-stabilized form of the original Wuhan SARS-CoV2 isolate, mice developed potent nAb responses against infectious SARS-CoV2 (*McKay et al., 2020*). Similarly, a DNA-launched Semliki-forest virus replicon expressing the Wuhan SARS-CoV2 isolate spike-induced humoral and cellular immunity against SARS-CoV2 in mice (*Szurgot et al., 2021*). We have previously evaluated the LION/repRNA vaccine expressing the native A.1 spike in mice and pigtail macaques and found that vaccination-induced potent nAb activity (*Erasmus et al., 2020a*). Challenge studies showed vaccinated pigtail macaques were protected against SARS-CoV2 challenge (manuscript in preparation). However, these studies did not evaluate the ability of these vaccines to protect against emerging VoCs and we therefore sought to determine the ability of LION/repRNA to confer broad protection against heterologous SARS-CoV2 challenge.

In both mice and hamsters, we found distinct neutralizing responses when comparing homologous to heterologous infectious virus. Relative to homologous vaccination and challenge virus, the greatest decrease in neutralization activity was seen when mice or hamsters were vaccinated with A.1 or B.1.1.7 repRNAs and nAbs measured against B.1.351 virus or when animals were vaccinated with B.1.351 repRNA and nAbs measured against B.1.1.7 virus. In hamsters vaccinated with A.1 or B.1.1.7 repRNA, a >8-fold decrease in neutralization activity against the B.1.351 strain was found compared to hamsters vaccinated with the B.1.351 repRNA. These findings are consistent with previous studies showing reduced neutralization of convalescent and A.1-vaccinated sera against the B.1.351 and B.1.1.7 variants (*Shen et al., 2021a*; *Shen et al., 2021b*; *Wang et al., 2021*; *Zhou et al., 2021*) and provides new insights into the design of VoC-specific vaccines and their ability to provide cross-neutralizing immunity. Nevertheless, although we observed reduced neutralization activity against these heterologous variants, hamsters vaccinated with any of the three repRNAs were still significantly protected against viral replication in the lungs and lung pathology following heterologous SARS-CoV2 challenge. We also found significant reduction of viral shedding in the upper airway of even heterologous challenged hamsters, indicating that all vaccines induced immunity above the threshold likely necessary for protection from disease in hamsters.

Given the recent and alarmingly rapid spread of the B.1.617.2 (Delta) VoC, we also evaluated the ability of sera from these vaccinated hamsters to cross-neutralize B.1.617.2 and compare vaccine resistance to other VoCs. Similar to recent reports (*Edara et al., 2021*; *Planas et al., 2021*), the B.1.617.2 VoC exhibited a 6-fold reduction in nAb titers in A.1-vaccinated hamsters, and B.1.351 VoC still remained the most resistant to A.1 vaccination. Interestingly, B.1.1.7 vaccination induced robust cross-neutralization of B.1.617.2 with only a 3.5-fold reduction in nAb titers relative to homologous neutralization. Vaccination with B.1.351, however, did not provide very robust cross-neutralization of B.1.617.2, with a 10-fold reduction in nAb titers relative to homologous neutralization, suggesting that those previously infected with B.1.351 may remain susceptible to re-infection with the B.1.617.2 VoC.

While protection from SARS-CoV2 infection in the lower respiratory tract is likely important for prevention of severe disease and death, reduction of shedding of virus from the upper respiratory tract is likely key for interruption of SARS-CoV2 transmission. Indeed, a recent study demonstrated that despite sterilizing protection in the lower airway of vaccinated hamsters challenged with

homologous virus, transmission was still observed upon co-housing with naïve animals (**Wu et al., 2021**). And while a significant reduction in upper airway viral load, as measured by Sg RNA qPCR, was observed in these vaccinated animals, an attempt to isolate infectious virus was not reported. Our data showed that VoC-matched vaccination (i.e. B.1.1.7- or B.1.351-repRNA) induced more rapid clearance of homologous virus with sterilizing protection in the oral cavity observed for majority of animals at the day 2 PI oral swab. And while we and others have demonstrated sterilizing protection in the lower airway of animals, protection from upper respiratory infection appears to be a higher bar as a measure of vaccine efficacy as several vaccine studies in animal models have shown reduced control of SARS-CoV2 replication in the upper respiratory tract. In ChAdOx1 nCoV-19-vaccinated hamsters, vaccination reduced shedding of SARS-CoV2 in B.1.1.7 challenged but not B.1.351 challenged hamsters, despite sterile protection in the lungs (**Fischer et al., 2021**). In ChAdOx1 nCov-19-vaccinated rhesus macaques, no differences in SARS-CoV2 RNA loads in nasal swabs between vaccinated or unvaccinated animals were observed despite significantly reduced viral loads in the lungs (**van Doremalen et al., 2020**). Intranasal delivery of the ChAd nCoV-19 vaccine may improve control of viral shedding as intranasal but not IM vaccination resulted in significantly reduced viral shedding (**Hassan et al., 2021**; **van Doremalen et al., 2021**). In contrast, rhesus macaques vaccinated with mRNA-1273 showed significantly reduced viral RNA in both bronchoalveolar-lavage (BAL) and nasal swabs (**Corbett et al., 2020**). Similarly, rhesus macaques vaccinated with Ad26 had significantly reduced upper and lower respiratory titers although across vaccine candidates evaluated, more robust protection was observed in the BAL than nasal swabs (**Mercado et al., 2020**). Of the vaccine efficacy studies have been conducted in hamsters (**Brocato et al., 2021**; **Fischer et al., 2021**; **Hörner et al., 2020**; **Meyer et al., 2021**; **Mohandas et al., 2021**; **Rauch et al., 2021**; **van der Lubbe et al., 2021**; **Wu et al., 2021**; **Yinda et al., 2021**; **Zhang et al., 2021**), those that evaluated infectious virus in upper airways, including unmodified mRNA (**Rauch et al., 2021**), measles-vectored (**Hörner et al., 2020**), and Ad26-vectored (**van der Lubbe et al., 2021**) approaches, were unable to induce sterilizing protection in the upper airway of a majority of animals while an inactivated rabies virus-vectored approach was able to do so (**Kurup et al., 2021**). An important consideration of the above referenced studies is that these were within the context of A.1 vaccination and matched A.1 viral challenge; therefore, continued evaluation of deployed vaccines and those under development against emergent VoCs is needed.

A limitation of our study is that we did not measure T-cell responses in vaccinated animals. However, we have previously shown that our A.1-specific repRNA could induce T-cell responses in mice and non-human primates (**Erasmus et al., 2020a**) indicating this vaccine can induce T-cell responses in vaccinated animals. Furthermore, nAbs have been shown to strongly correlate with vaccine-mediated protection from SARS-CoV-2 infection (**Khoury et al., 2021**) and our data showed that vaccination with our A.1- or B.1.1.7-specific repRNA could induce heterologous neutralizing activity even against the B.1.617.2 VoC. However, we cannot exclude the possibility that T-cell responses contribute to this protection nor that T-cell responses may contribute to vaccine-mediated protection against B.1.351 challenge in A.1- or B.1.1.7-specific repRNA-vaccinated animals as we saw diminished or absent neutralizing activity against B.1.351 in these animals.

Together our data demonstrate that LION/repRNA vaccines can induce broadly protective immunity against multiple SARS-CoV2 strains. Vaccination significantly reduced SARS-CoV2 viral burden in both the upper and lower respiratory tract and prevented development of SARS-CoV2 lung pathology even when challenged with heterologous SARS-CoV2 strains. Additionally, given the distinct heterologous nAb responses elicited by B.1.1.7- or B.1.351-specific repRNA, coupled with the observation that matched vaccination provides superior protection in the upper airway against homologous VoC challenge, a bivalent mixture consisting of these two VoC-specific vaccines may provide superior cross-neutralizing capacity than each monovalent vaccine alone. Our data also support a vaccine approach in which the vaccine is matched to a VoC to provide optimal immunity which may decrease the frequency of vaccine breakthrough-mediated transmission events, an important consideration for the rapid emergence of the B.1.529 (Omicron) VoC (**Ai et al., 2022**; **Carreño et al., 2022**; **Kuhlmann et al., 2022**). Furthermore, our vaccine platform is amenable to rapid updates to specifically target emergent VoCs and our data cumulatively support continued development of the LION/repRNA vaccine platform for COVID-19.

## Materials and methods

### Biosafety and ethics

All procedures with infectious SARS-CoV2 were conducted under high biocontainment conditions in accordance with established operating procedures approved by the Rocky Mountain Laboratories (RML) Institutional Biosafety Committee (IBC). Sample inactivation followed IBC approved protocols (*Haddock et al., 2021*). Animal experiments were approved by the corresponding institutional animal care and use committee and performed by experienced personnel under veterinary oversight. Mice were group-housed, maintained in specific pathogen-free conditions, and entered experiments at 6–8 weeks of age. Hamsters were group-housed in HEPA-filtered cage systems and acclimatized to high containment conditions prior to start of SARS-CoV2 challenge. They were provided with nesting material and food and water ad libitum.

### Viruses and cells

Viruses used for hamster challenge were as described previously (*Hansen et al., 2021*). For in vitro assays: Vero USAMRIID (a gift from Ralph Baric, UNC-Chapel Hill), VeroE6-TMPRSS2 (JCRB1819, JCRB Cell Bank, NIBIOHN), and Vero-hACE2-TMPRSS2 (a gift from Michael Diamond, Washington University) cells were cultured at 37°C in DMEM supplemented with 10% FBS, and 100 U/ml of penicillin-streptomycin. In addition, VeroE6-TMPRSS2 and Vero-hACE2-TMPRSS2 cells were cultured in the presence of 1 mg/ml G418 and 10 μg/ml puromycin, respectively. Cell line identity was not authenticated. The SARS-CoV-2 Isolate hCoV-19/Germany/BavPat1/2020 (B.1) (NR-52370, BEI Resources), hCoV-19/England/204820464/2020 (B.1.1.7) (NR-54000, BEI Resources), and hCoV-19/USA/PHC658/2021 (B.1.617.2) (NR-55612, BEI Resources) were obtained from BEI Resources. Virus stocks were generated by expanding the virus in Vero-USAMRIID cells. Isolate 501Y.V2.HV001 (B.1.351) was obtained from Alex Sigal, AHRI (African Health Research Institute) and amplified in Vero-hACE2-TMPRSS2 cells upon reception. Virus stocks generated were tittered on VeroE6-TMPRSS2 cells. Viral RNAs were purified from the stock virus by using Quick-RNA viral kit (Zymo Research) and sent for RNA-seq for verification (University of Washington).

### Vaccine constructs

Spike variants were constructed on the background of either the KV995PP substitution to stabilize the pre-fusion conformation of spike, or the native spike without pre-fusion stabilizing substitutions. The full-length spike open reading frame derived from the original Washington isolate was used as the reference A.1 lineage spike, and the various deletions and substitutions that define the B.1, B.1.1.7, and B.1.351 lineages are depicted in *Figure 1*. All constructs were cloned by Gibson assembly of three overlapping fragments synthesized on a bioxp (Codex DNA) and codon optimized for human codon usage. Plasmids were then Sanger sequenced to confirm nucleotide identity and then linearized by NotI digestion prior to transcription and capping as described (*Erasmus et al., 2020b*).

### LION/repRNA potency assay

Serial dilutions of LION/repRNA were incubated on a monolayer of BHK cells in a 96-well plate. Twenty-four hours later, cell lysates were added to an ELISA plate coated with anti-SARS-CoV2 Spike (S1 domain) monoclonal antibody. Following a primary incubation and washes, a polyclonal anti-SARS-CoV2 Spike (full-length S) primary antibody was added. Following a secondary incubation and washes, a secondary horse radish peroxidase (HRP)-conjugated antibody was used to detect S-specific binding. Following a final incubation, HRP activity was assayed by TMB/HCL detection and absorbance measured by plate reader (EL$_x$808, Bio-Tek Instruments Inc) at 450 nm.

### Mouse studies

For mouse studies, 6- to 8-week-old female C57BL/6 mice (Jackson laboratory) received 1 μg of each vaccine, as outlined in *Table 1*, via IM injections on days 0 and 28. Animals were then bled 2 weeks after the booster immunization and sera was evaluated for nAb responses by plaque reduction neutralization test against A.1, B.1, B.1.1.7, and B.1.351 viruses.

### Hamster studies

For hamster studies, 7- to 8-week-old male Syrian Golden hamsters were purchased from Envigo. Hamsters were randomly assigned to study groups and acclimatized for several days prior to

vaccination. Hamsters were vaccinated with 20 μg of indicated repRNA complexed to LION. RNA was diluted in water and LION diluted in 40% sucrose and 100 mM sodium citrate to achieve a theoretical nitrogen:phosphate (N:P) ratio of 15. RNA and LION were allowed to complex for 30 min at 4°C. Hamsters were primed with a 50 μl IM injection to each hind limb on day 0 and boosted 4 weeks later. Mock-vaccinated hamsters received identical IM immunizations with saline. To monitor antibody responses to vaccination, blood was collected via retroorbital bleeds 25 days after prime vaccination and 14 and 21 days after boost vaccination. Hamsters were monitored daily for appetite, activity, and weight loss, and no adverse events were observed among the LION/repRNA-vaccinated groups. Following the first vaccination, one mock-vaccinated hamster developed a testicular abscess and per veterinarian recommendation was euthanized. Data from this hamster was excluded from all analyses. For SARS-CoV2 challenge, hamsters were inoculated with 1000 $TCID_{50}$ indicated SARS-CoV2 variant via 50 μl intranasal instillation. Following challenge, hamsters were weighed and monitored daily. Hamsters were orally swabbed on days 2 and 4 PI. Swabs were placed in 1 ml DMEM without additives. A scheduled necropsy at day 4 PI was performed on all animals to harvest blood and lung tissue. Studies were performed once.

## Viral RNA quantification

Viral RNA from swabs was isolated using Qiamp RNA mini kit (Qiagen) and viral RNA was isolated from tissues using RNEasy mini kit (Qiagen) according to provided protocols. Viral RNA was quantified by one-step qRT-PCR using QuantiFast Probe PCR reagents (Qiagen) and primers and probes specific for the SARS-CoV2 Sg E RNA as previously described (*Corman et al., 2020*). For both assays, cycling conditions were as follows: initial hold of 50°C for 10 min, initial denaturation of 95°C for 5 min, and 40 cycles of 95°C for 15 s followed by 60°C 30 s. SARS-CoV2 RNA standards with known copy number were prepared in house, diluted, and run alongside samples for quantification. The limit of detection was based on the standard curve and defined as the quantity of RNA that would give a Ct value of 40.

## Infectious virus titration

Infectious virus in swabs or tissues was quantified by $TCID_{50}$ on Vero cells. Tissues were weighed and homogenized in 1 ml DMEM supplemented with 2% FBS and penicillin and streptomycin. Homogenate was clarified of large debris by centrifugation. Samples were then serially 10-fold diluted in DMEM 2% FBS and applied to wells beginning with the 1:10 dilution in triplicate. Cells were incubated for 6 days before cytopathic effect (CPE) was read. $TCID_{50}$ was determined by the Reed and Muench method (*Reed and Muench, 1938*). The limit of detection was defined as at least two wells positive in the 1:10 dilution. To distinguish samples with no detectable infectious virus from those with single positive wells (<1 median TCID), *Table 1* reports the fraction of samples from each group with any wells positive for CPE.

## Plaque reduction neutralization tests

Two-fold serial dilutions of heat-inactivated serum and 600 plaque-forming units (PFU)/ml solution of A.1, B.1, B.1.1.7, B.1.351, or B.1.617.2 viruses were mixed 1:1 in DMEM and incubated for 30 min at 37°C. Serum/virus mixtures were added, along with virus-only and mock controls, to Vero E6-TMPRSS2 cells (ATCC) in 12-well plates and incubated for 30 min at 37°C. Following adsorption, plates were overlayed with a 0.2% agarose DMEM solution supplemented with penicillin/streptomycin (Fisher Scientific). Plates were then incubated for 2 days at 37°C. Following incubation, 10% formaldehyde (Sigma-Aldrich) in DPBS was added to cells and incubated for 30 min at room temperature. Plates were then stained with 1% crystal violet (Sigma-Aldrich) in 20% EtOH (Fisher Scientific). Plaques were enumerated and percent neutralization was calculated relative to the virus-only control.

## Histology and IHC

At time of necropsy, lungs were dissected and insufflated with 10% neutral buffered formalin and then submerged in 10% neutral buffered formalin for a minimum of 7 days with two changes. Tissues were placed in cassettes and processed with a Sakura VIP-6 Tissue Tek, on a 12 hr automated schedule, using a graded series of ethanol, xylene, and ParaPlast Extra. Prior to staining, embedded tissues were sectioned at 5 μm and dried overnight at 42°C. Using GenScript U864YFA140-4/CB2093 NP-1 (1:1000) specific anti-CoV immunoreactivity was detected using the Vector Laboratories ImPress VR

anti-rabbit IgG polymer (#MP-6401) as secondary antibody. The tissues were then processed using the Discovery Ultra automated processor (Ventana Medical Systems) with a ChromoMap DAB kit Roche Tissue Diagnostics (#760–159). Sections were scored by certified pathologists who were blinded to study groups.

## Statistical analyses

Statistical analyses as described in the figure legends were performed using Prism v9 (GraphPad).

## Acknowledgements

The authors thank the Rocky Mountain Veterinary Branch (NIAID, NIH) for animal care and husbandry and the Research Technologies Branch (NIAID, NIH) for sequencing of stock viruses. We are grateful to Emmie de Wit and Vincent Munster for their discussions and help with virus stock preparations. The B.1.1.7 variant was obtained through BEI Resources (Bassam Hallis, Sujatha Rashid), NIAID (Ranjan Mukul, Kimberly Stemple), and the NIH. This research was supported in part by the Intramural Research Program, NIAID/NIH, and by grants 27220140006C (JHE), AI100625, AI151698, and AI145296 (MG). Funders had no role in study design, data interpretation, or decision to publish.

## Additional information

### Competing interests

Jacob Archer, Michael Gale: has equity interest in HDT Bio. Amit P Khandhar: has equity interest in HDT Bio. Is a co-inventors on U.S. patent application no. 62/993,307 "Compositions and methods for delivery of RNA" pertaining to the LION formulation. Peter Berglund: has equity interest in HDT Bio. Is a consultant for Arcturus, Sensei, and Next Phase. Deborah Heydenburg Fuller: has equity interest in HDT Bio. Is a consultant for Gerson Lehrman Group, Orlance, Abacus Bioscience, Neoleukin Therapeutics. Jesse H Erasmus: has equity interest in HDT Bio. Is a consultant for InBios. Is a co-inventors on U.S. patent application no. 62/993,307 "Compositions and methods for delivery of RNA" pertaining to the LION formulation. The other authors declare that no competing interests exist.

### Funding

| Funder | Grant reference number | Author |
|---|---|---|
| National Institute of Allergy and Infectious Diseases | 27220140006C | Michael Gale Jesse Erasmus |
| Division of Intramural Research, National Institute of Allergy and Infectious Diseases | | David W Hawman Kimberly Meade-White Shanna S Leventhal Drew Wilson Carl Shaia Heinz Feldmann |
| National Institute of Allergy and Infectious Diseases | AI100625 | Michael Gale Jesse H Erasmus |
| National Institute of Allergy and Infectious Diseases | AI151698 | Michael Gale Jesse H Erasmus |
| National Institute of Allergy and Infectious Diseases | AI145296 | Michael Gale Jesse H Erasmus |

The funders had no role in study design, data collection and interpretation, or the decision to submit the work for publication.

### Author contributions

David W Hawman, Conceptualization, Data curation, Formal analysis, Investigation, Methodology, Project administration, Supervision, Visualization, Writing - original draft, Writing – review and editing; Kimberly Meade-White, Data curation, Investigation, Methodology; Jacob Archer, Samantha Randall, Amit P Khandhar, Peter Berglund, Conceptualization, Resources; Shanna S Leventhal, Formal analysis,

Investigation, Methodology; Drew Wilson, Investigation; Carl Shaia, Kyle Krieger, Formal analysis, Investigation; Tien-Ying Hsiang, Resources; Michael Gale, Conceptualization, Supervision; Deborah Heydenburg Fuller, Conceptualization, Resources, Writing – review and editing; Heinz Feldmann, Conceptualization, Funding acquisition, Project administration, Resources, Supervision, Writing – review and editing; Jesse H Erasmus, Conceptualization, Formal analysis, Methodology, Resources, Writing – review and editing

#### Author ORCIDs
David W Hawman http://orcid.org/0000-0001-8233-8176
Heinz Feldmann http://orcid.org/0000-0001-9448-8227
Jesse H Erasmus http://orcid.org/0000-0003-1612-2697

#### Ethics
Animal experiments were approved by the corresponding institutional animal care and use committee and performed by experienced personnel under veterinary oversight (Protocol #2020-63).

#### Decision letter and Author response
Decision letter https://doi.org/10.7554/eLife.75537.sa1
Author response https://doi.org/10.7554/eLife.75537.sa2

## Additional files

### Supplementary files
• Supplementary file 1. Complete histological and immunohistochemistry (IHC) findings. Formalin-fixed lung tissue was stained with H&E or viral antigen detected via IHC. Sections were scored by a certified pathologist blinded to study groups. Representative images are shown in *Figure 6*, scores in *Figure 7*, and complete summary of findings are provided here.

• Transparent reporting form

### Data availability
All data generated or analyzed during this study are included in the figures and supporting files.

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
