## [Editor Report]

This manuscript aims to develop second-generation vaccines that protect against multiple SARS-CoV2 variants. The overall experimental design and the data are very nice. In addition, authors reasonably revised the original manuscript.

---

## [Decision Letter]

**Decision letter after peer review:**

Thank you for submitting your article "SARS-CoV2 variant-specific replicating RNA vaccines protect from disease and pathology and reduce viral shedding following challenge with heterologous SARS-CoV2 variants of concern" for consideration by *eLife*. Your article has been reviewed by 2 peer reviewers, and the evaluation has been overseen by a Reviewing Editor and Prof. Sara Sawyer as the Senior Editor. The reviewers have opted to remain anonymous.

This study is aimed to make broadly effective vaccines against possible mutants of SARS-CoV2. As seen in the below reviewers' comments, both are positive for this manuscript, although they have some concerns, which should be addressed.

Essential revisions:

1) Infection protection against B1.617.2 (δ strain) is important, as mentioned in the reviewer's 2 public review (1).

2) Examining T cell responses is also important after vaccination (mentioned in the reviewer's 1 public review (3) and the reviewer's 2 public review (2).

*Reviewer #1 (Recommendations for the authors):*

The present manuscript has low novelty. To better understand the great efficacy of this vaccine for SARS-CoV2 variant and provide the novelty of this manuscript, it will be better to add some data for cross-neutralization activity against "omicron" in the manuscript.

*Reviewer #2 (Recommendations for the authors):*

1) The behavior of each PRNT80 reciprocal titer in Figures 2A and 2B is largely different. Why are there so many individual differences among mice of the same strain? Also, the authors should add the median values to the graph.

2) The authors should discuss that A.1 virus shedding does not decrease in hamsters vaccinated with the A.1 vaccine, but a different strain, B.1.1.7, does.

3) There is a sentence "Figure 5C, F, I" in the text, but it does not match the figure. Please correct the sentence.

4) The same diagram is used for A.1-challenge/A.1-vaccine and A.1-challenge/B.1.351-vaccine in Figure 6. Please correct the figures.

---

## [Author Response]

Reviewer #1 (Recommendations for the authors):The present manuscript has low novelty. To better understand the great efficacy of this vaccine for SARS-CoV2 variant and provide the novelty of this manuscript, it will be better to add some data for cross-neutralization activity against "omicron" in the manuscript.

We thank the reviewer for the suggestions and comments on the paper. However, we consider our efficacy data to be novel and relevant to readers by showing that although cross-protection is achievable with our vaccine, mismatches between the vaccine antigen and SARS-CoV-2 variant can lead to diminished immunity and protection from infection, particularly in the upper airway. These data would support matching a vaccine to the circulating VoC to provide optimal immunity and potentially enhance the transmission blocking capacity of vaccination. We have included a revised figure 1, accompanying Results section and discussion to emphasize our capacity to modify and validate our vaccine platform to target VoCs as they emerge. This is an important and relevant consideration considering the explosion of omicron even among vaccinated populations. Efficacy testing of this vaccine against the omicron variant certainly is an important question. We have already started the omicron-specific project but it is still ongoing and falls outside the scope of this manuscript. It is likely that once we conclude the omicron project there will yet again be another VoC to chase; it would be unfeasible to continue to update a manuscript with the latest VoC-specific data. We have included updated conclusion statements to emphasize these conclusions.

Reviewer #2 (Recommendations for the authors):1) The behavior of each PRNT80 reciprocal titer in Figures 2A and 2B is largely different. Why are there so many individual differences among mice of the same strain? Also, the authors should add the median values to the graph.

We have updated figure 2 to show geometric mean values, included data on the stabilized versus native conformation spike vaccinations and clarified the Results section to hopefully make the data more clear. The reason for variation among vaccinated mice receiving the same vaccine is unknown but could be due to stochastic events in either the animals or assay. Nevertheless, we included sufficient biological replicates within the studies to account for this variability.

2) The authors should discuss that A.1 virus shedding does not decrease in hamsters vaccinated with the A.1 vaccine, but a different strain, B.1.1.7, does.

We appreciate the suggestion and have updated the text in the results to describe this finding, including additional data that correlate this finding with an increase in S1-binding antibody responses in the A.1 spike. We do not have a definitive explanation but speculate that the differences may either be stochastic as we observed no difference in neutralizing activity against A.1 when hamsters were vaccinated with either A.1 or B.1.1.7-specific vaccines, or due to higher S1-binding responses.

3) There is a sentence "Figure 5C, F, I" in the text, but it does not match the figure. Please correct the sentence.

We have corrected the manuscript.

4) The same diagram is used for A.1-challenge/A.1-vaccine and A.1-challenge/B.1.351-vaccine in Figure 6. Please correct the figures.

We thank the reviewer for catching this error and have corrected the figure.